# Vision-Language Modelling for Radiological Imaging and Reports in the Low Data Regime

**Rhydian Windsor**[1]                                           RHYDIAN@ROBOTS.OX.AC.UK
[1] *Visual Geometry Group, Department of Engineering Science, University of Oxford*

**Amir Jamaludin**[1]                                            AMIRJ@ROBOTS.OX.AC.UK

**Timor Kadir**[1]

**Andrew Zisserman**[1]

**Editors:** Accepted for publication at MIDL 2023

## Abstract

This paper explores training medical vision-language models (VLMs) – where the visual and language inputs are embedded into a common space – with a particular focus on scenarios where training data is limited, as is often the case in clinical datasets. We explore several candidate methods to improve low-data performance, including: (i) adapting generic pre-trained models to novel image and text domains (i.e. medical imaging and reports) via unimodal self-supervision; (ii) using local (e.g. GLoRIA) & global (e.g. InfoNCE) contrastive loss functions as well as a combination of the two; (iii) extra supervision during VLM training, via: (a) image- and text-only self-supervision, and (b) creating additional positive image-text pairs for training through augmentation and nearest-neighbour search.

Using text-to-image retrieval as a benchmark, we evaluate the performance of these methods with variable sized training datasets of paired chest X-rays and radiological reports. Combined, they significantly improve retrieval compared to fine-tuning CLIP, roughly equivalent to training with $10\times$ the data. A similar pattern is found in the downstream task classification of CXR-related conditions with our method outperforming CLIP and also BioVIL, a strong CXR VLM benchmark, in the zero-shot and linear probing settings. We conclude with a set of recommendations for researchers aiming to train vision-language models on other medical imaging modalities when training data is scarce. To facilitate further research, we make our code publicly available[1].

**Keywords:** Vision-Language Modelling, Contrastive Learning, Chest X-ray

## 1. Introduction

Recently, there has been much progress in the field of vision-language modelling (VLM) where powerful, aligned, visual and language representations, such as CLIP and ALIGN, are learnt from image-caption pairs scraped from the internet (Radford et al., 2021; Jia et al., 2021; Ilharco et al., 2021). This has exciting implications for deep learning in medical imaging, where annotating large datasets for supervised training has been a long-standing challenge since it generally requires an expert annotator whose time is expensive and limited. Fortunately, almost all scans taken in a clinical setting will have a corresponding radiological

---

1. https://github.com/rwindsor1/Data-Efficient-CXR-VLM

report, describing key findings in free-text. VLMs can use these reports as a supervisory signal to learn representations of images, forgoing the need for manual annotation. However, popular general-domain VLM methods use incredibly large datasets of paired images and text – CLIP, for example, is trained on 400 million image-text pairs. Clearly, this is not feasible in the medical domain, where dataset size is limited by factors such as scanner availability, concerns about patient privacy and the cost of obtaining images. For some types of imaging investigations, researchers can realistically hope for no more than a few thousand training image-text pairs, even if data is collected from multiple imaging centres.

To alleviate this problem, we investigate a set of methods to improve VLM performance for dual encoder models when the number of image-text pairs available for training is limited. We begin by proposing several candidate methods to achieve this (Section 2). We evaluate these methods by using them to train models on progressively smaller datasets of paired chest X-rays and reports (Section 3). The aim is to find methods with minimal degradation in performance when the training set size is shrunk, as measured by retrieval on paired image-text data unseen during training. We show that with appropriate pretraining and supervision, a dual-encoder trained on thousands of paired image-text samples can achieve a performance comparable to one trained on hundreds of thousands of paired samples. Having identified the best performing training methods for retrieval, we then explore whether the improved retrieval corresponds to better performance on downstream tasks, namely zero-shot classification of common conditions associated with chest X-rays. We conclude by offering a set of recommendations for researchers aiming to train VLMs on novel medical imaging domains with limited data (Section 4).

## 1.1. Related Work

In this work, we target dual encoder VLMs, that learn a joint visual-language representation, rather than generative models that ingest images and output text such as (Alayrac et al., 2022) for generic images and text or (You et al., 2021; Nooralahzadeh et al., 2021; Yang et al., 2022; Kayser et al., 2022) for automatic medical image report generation.

Several works have been published relating to learning joint representations of medical images and reports, mostly in the domain of chest X-rays (CXRs) – This is likely due to the existence of multiple large-scale, publicly available CXR datasets such as MIMIC-CXR (Johnson et al., 2019), CheXpert (Irvin et al., 2019) and PadChest (Bustos et al., 2020). The majority of these works aim to learn representations of images and their associated reports using a contrastive learning paradigm (i.e. matching associated image-text pairs together across a randomly sampled batch) (Wang et al., 2018; Huang et al., 2021; Liao et al., 2021; Müller et al., 2022; Boecking et al., 2022; Zhang et al., 2022). Several works have demonstrated that strong performance at this pre-training task correlates with performance across a wide range of downstream tasks, such as classification, segmentation, and natural language inference tasks. While many of these works explore performance when data for downstream tasks is limited, the setting of limited data for VLM training remains relatively neglected. This is a particularly important problem for medical imaging since few modalities other than CXR have publicly-available datasets of 100,000s image-text pairs. A few recent works (Segal et al., 2022; Li et al., 2022; Mu et al., 2021) reduced bimodal training data

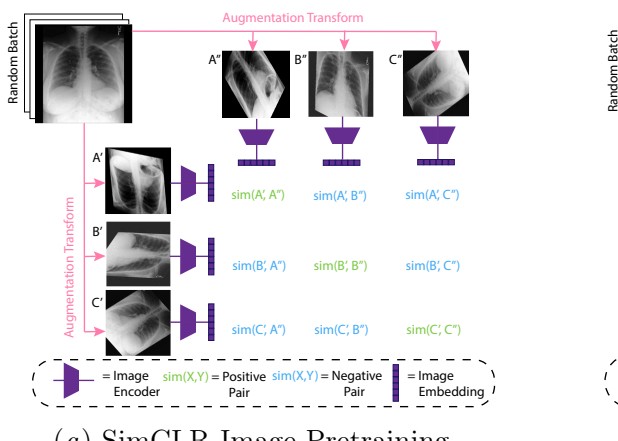
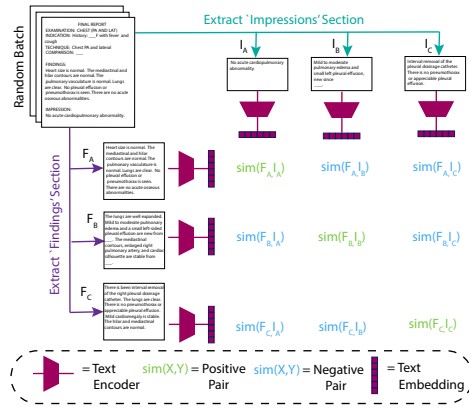

(*a*) SimCLR Image Pretraining  (*b*) CXR-BERT Specialization

**Figure 1:** The unimodal pretraining methods used to domain adapt image and text encoders before VLM training.

for generic image-text pairs. However, these works consider subsets of exceptionally large datasets (of the order of millions of image-text pairs), rather than 1000s as considered here.

## 2. Increasing Data Efficiency During Vision-Language Modelling

In this section, we outline possible methods for achieving strong dual encoder VLM performance when a limited number of *paired* image-text data are available for training. However, as is often the case, a large number of *unpaired* (unimodal) data may well be available, and we will take advantage of this to adapt the image and text encoders to the target domains before paired training begins. Broadly, the methods can be separated into two stages: (i) choosing an optimal dual encoder initialization by using image and text models trained to perform related unimodal self-supervised tasks; and (ii) adding more supervision during paired (bimodal) training by varying the contrastive loss function and also by generating additional positive image-text pairs. These two stages are described below:

**Stage 1: Domain-adapting image and text encoders:** In this stage, we start with strong, generic image and text encoders (e.g. an ImageNet-pretrained image model, and a BERT style language model pre-trained on large corpora). The encoders are then adapted to the CXR domain by self-supervised unimodal training as follows:

*Image Encoder:* To pretrain the image encoder, unimodal domain adaption is achieved using SimCLR (Chen et al., 2020). In this method, multiple views of each image in the dataset are created using a range of augmentations. The image encoder is then trained contrastively such that embedding vectors corresponding to different views of the same original image should have high cosine similarity. This process is illustrated in Figure 1(*a*).

*Text Encoder:* To train a domain-adapted text encoder, we follow the suggestions of (Boecking et al., 2022). We begin with CXR-BERT, a BERT model (Devlin et al., 2018) and tokenizer trained using standard RoBERTa-style masked language modelling (MLM) on two large corpora of generic clinical text datasets (PubMed Abstracts and MIMIC clinical notes), with a comparatively small amount of domain-specific data added in (MIMIC-CXR

reports). The resulting model is then 'specialized' to chest X-ray reports by an additional pre-training step, whereby the language model to trained to contrastively match a report's 'findings' section to the 'impressions' section (i.e. summary) from the same report. Conceptually, this is analogous to SimCLR, except instead of different views of the same image, different sections from the same report (which should contain the same information) are matched together. This step is shown in Figure 1(*b*).

Note that both these methods allow potentially larger datasets of unpaired images or reports to be used in training the model, alleviating the problem of limited paired data.

**Stage 2a: Local vs. Global Loss Functions:** Standard VLM training methods such as CLIP and ALIGN represent each image and each caption as a single global embedding vector and attempt to maximise the similarity between vectors from the same pair, usually via the InfoNCE loss function (Oord et al., 2018):

$$\mathcal{L}_{InfoNCE} = -\frac{1}{2B}\sum_{i=0}^{B}\left[\log\frac{\exp(-\mathbf{v}_i.\mathbf{t}_i/\tau)}{\sum_j^B\exp(-\mathbf{v}_i.\mathbf{t}_j/\tau)} + \log\frac{\exp(-\mathbf{v}_i.\mathbf{t}_i/\tau)}{\sum_j^B\exp(-\mathbf{v}_j.\mathbf{t}_i/\tau)}\right]. \tag{1}$$

Here, $\mathbf{v}_i, \mathbf{t}_i \in \mathbb{R}^E$ represent $E$-dimensional L2-normalized embedding vectors of the image and text respectively for the $i$-th image-text pair in a batch of size $B$. $\tau$ represents the softmax temperature.

However, in medical imaging, reports often contain multiple unrelated statements, each referring to distinct regions within the image (e.g. 'left lung base atelectasis, mild cardiomegaly, right lung normal'). Since global embeddings are usually produced by a pooling operation across the whole image, maintaining this spatial information in the embeddings is challenging. A possible solution to this problem are *local representations*, whereby image patches and text tokens are each represented by their own embedding vector. The model is trained such that the mean maximum patch-to-token and token-to-patch similarity is high for matching image-text pairs and low otherwise. In this paper, we consider GLoRIA, the local loss function proposed in (Huang et al., 2021). Concretely, for a image-text pair with $W$ word embeddings, $\mathbf{t} \in \mathbb{R}^{W \times E}$ and local embedding vectors describing $M$ image regions, $\mathbf{v} \in \mathbb{R}^{M \times E}$ (again both L2-normalized), a similarity matrix $S \in \mathbb{R}^{W \times M}$ is calculated, where $s_{ij}$ represents the cosine similarity between the $i$-th word embedding and $j$-th image region embedding. Using, this an attention score is calculated,

$$a_{ij} = \frac{\exp(s_{ij}/\tau_2)}{\sum_{k=1}^{M}\exp(s_{ik}/\tau_2)}. \tag{2}$$

This is then used to calculate an attention-weighted representation of the image for word $i$, $\mathbf{c}_i = \sum_{j=0}^{M} a_{ij}\mathbf{v}_{ij} \in \mathbb{R}^E$. The total report-image similarity is given by the matching function

$$Z(\mathbf{v}, \mathbf{t}) = \log(\sum_{i=1}^{W}\exp(\mathbf{c}_i.\mathbf{t}_i/\tau_3)^{\tau_3}. \tag{3}$$

$Z(\mathbf{v}, \mathbf{t})$ then replaces $\mathbf{v}.\mathbf{t}$ in equation 1, giving the following symmetric loss for a batch:

$$\mathcal{L}_{GLoRIA} = -\frac{1}{2B}\sum_{k=1}^{B}\left[\log\frac{\exp(-Z(v,t)/\tau)}{\sum_l^B\exp(-Z(v_k,t_l)/\tau)} + \log\frac{\exp(-Z(v,t)/\tau)}{\sum_l^B\exp(-Z(v_l,t_k)/\tau)}\right]. \tag{4}$$

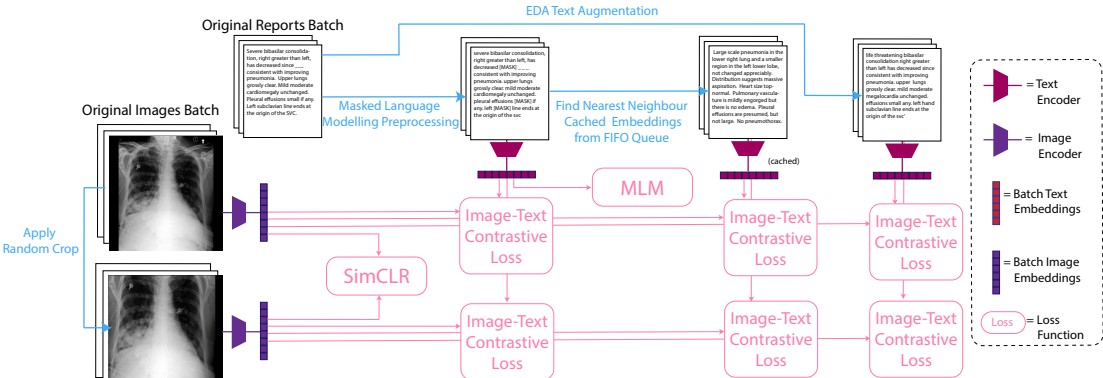

**Figure 2:** The DeCLIP framework used to add additional supervision during VLM training. Note that the image-text contrastive loss function can be either local or global.

As in the original GLoRIA paper, we also explore a simple combination of global and local loss functions, i.e. $\mathcal{L} = \mathcal{L}_{InfoNCE} + \mathcal{L}_{GLoRIA}$.

**Stage 2b: Additional Supervision During VLM Training:** As well as varying the form of the contrastive loss function, we investigate two further adaptions to VLM training. These are: (a) adding text-only & image-only self-supervision terms to the total loss; (b) creating more positive image-text pairs via augmentation and nearest-neighbour search. To do this, we use the Data efficient CLIP (DeCLIP) framework (Li et al., 2022). The overall process is illustrated in Figure 2. For image-only self-supervision, SimCLR is used (as in Stage 1). Text-only self-supervision is provided by masked-language modelling. Additional positive text-image pairs are generated as follows: Starting with a matching image-text pair, $\{\mathbf{v}_i, \mathbf{t}_i\}$, random-crop augmentations are used to generate another view of the image, $\mathbf{v}'_i$. Similarly, another 'view' of the text, $\mathbf{t}'_i$, is created by EDA (Wei and Zou, 2019) text augmentation, involving random synonym replacement, word insertion, deletion, and position swapping. Finally, a nearest neighbour to the text, $\mathbf{t}_i^{NN}$ is sampled from the data as follows: text embeddings are cached during training in a first-in, first-out (FIFO) queue. Once the queue is populated, the nearest neighbour for each text embedding is found by cosine similarity and used as a positive example for both corresponding images. This results in 6 positive pairs which can be used for contrastive training; $\{\mathbf{v}_i, \mathbf{t}_i\}$, $\{\mathbf{v}'_i, \mathbf{t}_i\}$, $\{\mathbf{v}_i, \mathbf{t}'_i\}$, $\{\mathbf{v}'_i, \mathbf{t}'_i\}$, $\{\mathbf{v}_i, \mathbf{t}_i^{NN}\}$ and $\{\mathbf{v}'_i, \mathbf{t}_i^{NN}\}$. This framework is agnostic to the choice of the contrastive loss function, and can be adapted for both local (e.g. GLoRIA) and global (e.g. InfoNCE) matching functions. For a more detailed explanation of this method and the hyper-parameters used, see Appendix C.

## 3. Experiments

**Dataset:** For the experiments in this paper, we use the MIMIC-CXR-JPG dataset. (Johnson et al., 2019). We include all reports with at least one frontal X-ray (AP or PA) and an 'Impression' section in the associated report, leaving 181,112 studies (144,781/18,679/17,652 in the training, validation, and test sets respectively). Each image is resized to 512×512 pixels by bilinear interpolation. For the reports, we used only the 'Impression' section as input to the text encoder during VLM training, however, the 'Findings' section is also used when domain-adapting the text encoder. We leave incorporating the longer 'Findings' sections

into VLM training as future work. The details of the training, testing  and validation splits used are given in Appendix A.

**Architecture, Evaluation metrics & Implementation details:**   In all experiments, we use a ResNet50 (He et al., 2016) image encoder and a BERT-style text encoder. All configurations are trained using an Adam optimizer with a learning rate of $10^{-3}$ and $\beta = (0.9, 0.99)$ with a batch size of 20 and softmax temperature of 0.5. Models are trained for 500 iterations and then evaluated for 100 iterations on the same subset of the validation set. Training continues until the validation loss does not improve for 10 successive validation epochs. Further details on each stage of training, including augmentations used, computational resources and training times are given in Appendix B. We evaluate models by measuring text-to-image retrieval on the test set after training has concluded. This involves calculating the similarity between each image and report in the test split and measuring true and false positive rates at a range of similarity thresholds. We report the area under the resulting ROC curve (AUROC) for models trained with 1, 5, 10, 20, 50, and 100% of the possible training pairs.

**Stage 1: Varying Dual Encoder Initialization:**   Figures $3(a)$ and $3(b)$ show results of experiments varying the weight initialization for both the image and text encoders, before training the dual-encoder using an InfoNCE contrastive loss. For the experiments varying image encoder pretrained weights, a standard uncased BERT base model is used to initialize the text encoder. We experiment with three image schemes: random initialization, standard ImageNet pretrained weights, and ImageNet weights with further domain adaption via image-only SimCLR pretraining. For the experiments varying the text encoder initialization, the SimCLR domain adapted image encoder is used. We compare four BERT models as text initializations: uncased BERT-base, ClinicalBERT, and CXR-BERT with and without further domain adaption. For both sets of experiments, we also plot the results of initializing the text and image encoders with ResNet50-CLIP pretrained weights. For both the images and text, initializing the encoder using weights from models trained via uni-modal self-supervised domain-adaption shows clear benefits. This is true at all dataset sizes, however, the effects are particularly pronounced with smaller training dataset sizes (1-20% of the data). For this reason, the SimCLR-domain adapted ResNet and domain-adapted CXR-BERT are used for initialization in all subsequent experiments unless otherwise stated.

**Stage 2a: Local vs. Global Loss Functions:**   Here, we explore varying the form of contrastive loss function used to match images and text. In total, three configurations are tested: (i) InfoNCE (equation 1); (ii) the local loss function component of GLoRIA (equation 4); and (iii) a combination of (i) and (ii). Note that for local loss functions, calculating pairwise similarities requires cross-attention between the image and text embedding vectors (see equations 2 and 3). This retrieval operation is generally more accurate however much slower and less suited to large scale datasets (see (Miech et al., 2021) for further discussion of this issue). In this paper, we report retrieval performance using both this local, cross attention mechanism (*local matching*, marked '(L)' in Figure 4) as well as via global embedding vector cosine similarity (*global matching*, marked '(G)'). For methods that have both global and local objectives, both matching functions are reported.

Figure $4(a)$ shows a comparison of the three configurations. As before, we plot the performance of CLIP initialization with an InfoNCE objective for comparison purposes. Our results show local matching results in better retrieval performance than global matching

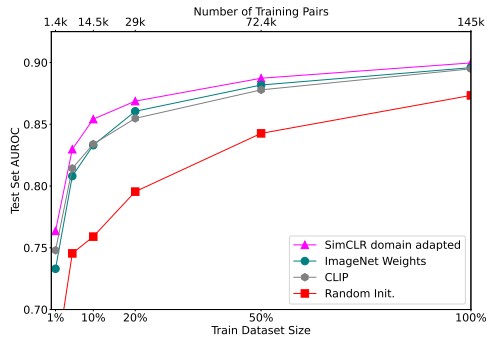 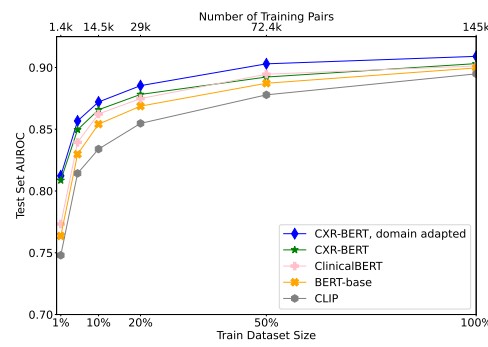

(*a*) Varying Image Encoder Initialization    (*b*) Varying Text Encoder Initialization

**Figure 3:** Training the dual encoder starting from various initializations for different size paired training data. (a) Varying the image encoder initializations, with a BERT-base-uncased as the initial text encoder. (b) Varying the text encoder initializations, with a simCLR-domain adapted ResNet50 model as the initial image encoder. InfoNCE contrastive loss is used to train the VLM, as in CLIP.

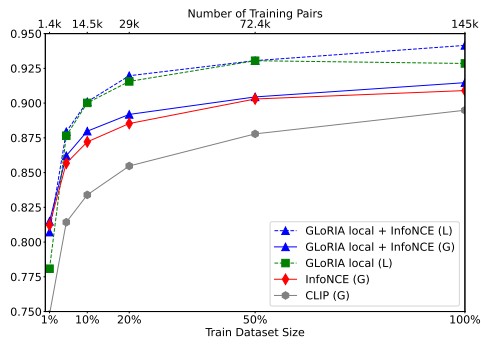 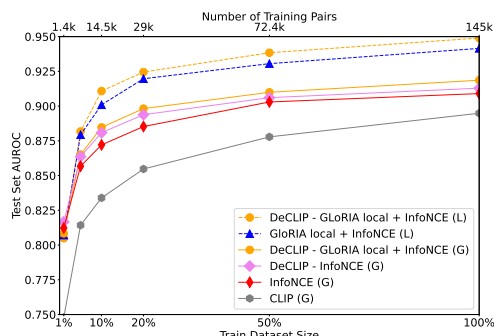

(*a*) Local vs. Global Loss Functions    (*b*) DeCLIP vs. standard VLM training

**Figure 4:** Varying the contrastive loss function used during VLM training. (a) Evaluates various local and global loss functions (b) DeCLIP evaluated against the standard InfoNCE approach. Methods performing retrieval using a local matching function (`L`) are shown as dashed lines while retrieval using a global matching function (`G`) are shown as solid lines. In all cases, the simCLR domain-adapted ResNet50 and domain-adapted CXR-BERT from stage 1 are used for initialization.

at all except very small training dataset sizes (1%). Crucially, we find that the extra supervision provided by both a local and global component to the contrastive loss function (`GLoRIA local + InfoNCE`), improves retrieval at both global and local matching, compared to respective methods which use a local (`GLoRIA local`) or global (`InfoNCE`) loss alone.

**Stage 2b: Additional Supervision via DeCLIP:** Here, we investigate the effect of adding more supervision via the DeCLIP framework, shown in Figure 2. This is done using two forms of contrastive loss function: (a) InfoNCE alone; (b) InfoNCE and the local component of GLoRIA. The trained models are compared to those trained by equivalent methods, but without the extra supervision given by DeCLIP. The results of this experiment

| Dataset Size | | % Average Balanced Accuracy | | | |
| --- | --- | --- | --- | --- | --- |
| | | Zero-Shot | | Linear Probe | |
| Frac. | # Pairs | CLIP init. | Ours | CLIP init. | Ours |
| 1% | 1,448 | 60.8 | 65.7 | 67.4 | 70.1 |
| 5% | 7,239 | 64.4 | 66.2 | 69.5 | 71.2 |
| 10% | 14,478 | 64.6 | 66.4 | 69.9 | 71.9 |
| 20% | 28,956 | 66.7 | 66.7 | 70.8 | 72.8 |
| 50% | 72,390 | 65.7 | 68.7 | 71.9 | 73.7 |
| 100% | 144,781 | 65.3 | 68.5 | 72.6 | 74.5 |

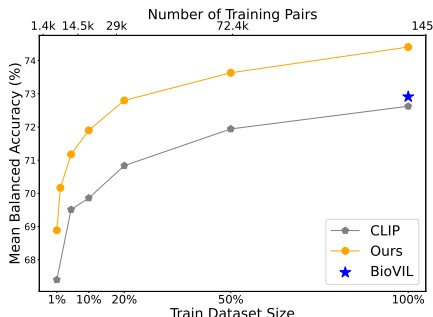

**Figure 5:** Zero-shot classification and linear probing of common conditions using CheXpert labels on the test split. For comparison, BioVIL achieves balanced accuracy scores of 67.3% and 72.9% in the zero-shot and linear probe settings respectively.

are shown in Figure 4(*b*). Encouragingly, DeCLIP improves retrieval performance in all settings at almost all dataset sizes. This is also additive with the improvement found by using a global and local loss function shown in Figure 4(*a*). An ablation study of each component of DeCLIP is shown in Appendix C, along with example nearest-neighbour reports.

Overall, with the benefits of self-supervised pretraining (Stage 1) and the combination of global and local loss functions with additional DeCLIP supervision (Stage 2), the retrieval curve rises quickly in the low-data regime, approaching a horizontal asymptote. For example, for 10% of the training data, the performance achieves over 95% of the final value (using 100% of the data). Also, the combination of methods far exceeds the fine-tuned CLIP baseline. Example retrieval results for this model are shown in Appendix B.

**Evaluation on Downstream Tasks:** The experiments above suggest that several methods can be employed to improve global text-to-image retrieval on the test dataset. Here, we explore how these improvements correlate to downstream classification tasks. Specifically, classifying 12 common CXR-related conditions using labels from the CheXpert labeller (Irvin et al., 2019). The full details of this experimental set-up, as well as class-level results, are given in Appendix D. The results are shown in Figure 5. We evaluate our best retrieval model (`DeCLIP - InfoNCE + GloRIA local`) at zero-shot image classification and also under linear probing, with the results shown in Figure 5. For comparison, we report the performance of the CLIP-initialized models trained using InfoNCE alone and also BioVIL (Boecking et al., 2022), a strong, publicly-available VLM for CXRs which is also trained on MIMIC-CXR[2]. From Figure 5, one can see that the improvements in retrieval carry over to the downstream evaluation, with our method achieving better performance than CLIP models trained on 5-10× the data in the linear probing setting (e.g. ours trained with 1% achieves 70.1% accuracy compared to 69.9% for CLIP trained on 10%) The zero-shot results are noisier, however again our method improves on CLIP and BioVIL at all dataset sizes.

---

2. The original paper does not report a train-test split, thus BioVIL may be trained on some image-text pairs which appear in our test set.

## 4. Conclusion

This paper has explored several methods of improving VLM-models when image-text pairs are limited, a particularly relevant problem to medical imaging. Based on these results, we make the following recommendations: (a) unimodal, self-supervised 'domain-adaption' of generic image and text models before VLM training vastly improves retrieval performance at all dataset sizes; (b) a combination of global and local contrastive loss functions is also beneficial; (c) adding more supervision during VLM training, via uni-modal self-supervision and generated additional positive image-text pairs, is another method to increase performance. Combining these ideas, we achieve state-of-the-art performance at zero-shot classification on MIMIC-CXR-JPG when trained on the whole dataset, and maintain strong performance even with limited training data. We hope these results will be useful to researchers aiming to train VLMs on medical imaging domains, especially when training data is scarce. To aid further research in this area, our code and models will be made publicly available.

## Acknowledgments

We are grateful to our funders: Rhydian Windsor is supported by Cancer Research UK via the EPSRC AIMS CDT (EP/S024050/1). Amir Jamaludin and Andrew Zisserman are supported by EPSRC Programme Grant Visual AI (EP/T025872/1).

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

## Appendix A. Dataset Details

In this paper, we used the MIMIC-CXR-JPG dataset for training and testing. This is a large dataset of chest X-rays and associated radiological reports. DICOMs from the original MIMIC-CXR dataset are converted into JPG files[3] and the CheXpert labeller (Irvin et al., 2019) is used to extract labels from the reports. The dataset is split into training, validation and testing splits on the patient-level, as shown in Table 1.

| Dataset Split | # Unique Subjects | # Report-Scan Pairs | Folders |
|---------------|-------------------|---------------------|---------|
| Total | 65,398 | 180,650 | p10-p19 |
| Train | 52,428 | 144,319 | p12-p19 |
| Validation | 6,572 | 18,679 | p11 |
| Test | 6,398 | 17,652 | p10 |

**Table 1:** The splits used for MIMIC-CXR-JPG in the experiments in this paper.

## Appendix B. Additional Training Details

### B.1. Augmentation Hyperparameters

Table 2 reports augmentation hyperparameters for all training stages used in this paper. We report the augmentations for both the image domain-adaption stage using simCLR and for full VLM training. Note that the DeCLIP framework introduces an additional image augmentations by producing an augmented version of the original image for a simCLR-like unimodal supervision objective. In this case we simply use a 50% random crop of the already-augmented original image, maintaining the same aspect ratio.

### B.2. Training Time & Computational Resources

Most models are trained with a batch size of 20 using 2× 24GB NVIDIA Tesla P40 GPUs. VLM training using the DeCLIP framework requires an additional GPU of the same description to maintain the same batch size. Approximate training times for each stage are given in Table 3.

---

3. Further explanation of the MIMIC-CXR-JPG dataset is given at https://physionet.org/content/mimic-cxr-jpg/2.0.0/

| | VLM Training | SimCLR Image Pretraining |
|---|---|---|
| Translation | $\pm 5$px | $\pm 20$ |
| Rotation | $-10$ to $10°$ | $-180$ to $180°$ |
| Brightness | $\pm 20\%$ | $\pm 20\%$ |
| Shear ($x$-axis) | - | $40°$ |
| Scaling | $\pm 10\%$ | $\pm 10\%$ |
| Horizontal Flip | - | $p = 0.5$ |
| Gaussian Noise | $\pm 5\%, p = 0.5$ | $\pm 5\%, p = 0.5$ |
| Gaussian Blur | - | $\sigma = (1, 3, 5), p = 0.5$ |

**Table 2:** Image augmentation parameters used during VLM training and while domain adapting the image encoder using simCLR.

| Training Stage | Additional Information | # Samples | Training Time (hours) |
|---|---|---|---|
| Image Encoder Domain Adaption | - | 144,319 | 12 |
| Text Encoder Domain Adaption | - | 144,319 | 4 |
| VLM Training (100%) | CLIP framework, InfoNCE loss | 144,319 | 9 |
| VLM Training (100%) | CLIP framework, Combined loss | 144,319 | 10 |
| VLM Training (100%) | DeCLIP, InfoNCE loss | 144,319 | 20 |
| VLM Training (1%) | DeCLIP, Combined loss | 1,443 | 2.5 |
| VLM Training (10%) | DeCLIP, Combined loss | 14,432 | 9 |
| VLM Training (100%) | DeCLIP, Combined loss | 144,319 | 36 |

**Table 3:** Approximate training times for each stage of our training pipeline. Note that for the image and text encoder domain adaption stages, the models are initialized using weights from generic pretraining (ImageNet & CXR-Bert respectively).

## Appendix C. DeCLIP Method, Nearest Neighbours & Ablation Study

Data Efficient CLIP (DeCLIP) (Li et al., 2022) is a method for training CLIP-like models aimed at maintaining strong performance with less image-text pairs. The main idea behind this method is to add several additional forms of supervision during VLM training, alongside the standard image-text matching contrastive InfoNCE loss. This additional supervision can be broadly seperated into three classes; uni-modal self-supervision, multi-view supervision and nearest-neighbour supervision. These additional loss terms can be seen in Figure 2 and are described individually below:

**Uni-Modal Self-Supervision:** One can add in any form of image-only or text-only self-supervision in addition to the contrastive image-text matching function. We opt for two common image and text objectives, namely SimCLR (Chen et al., 2020) using random crop augmentations and masked language modelling, normally used to train BERT-like language models from scratch. This also acts as a form of text augmentation during VLM training since each token is randomly masked out or replaced with another random tokens with some small probability, $p$.

**Multi-View Supervision:** This component of the framework aims to add additional supervision by creating additional 'views' of the data; augmentations of the images, $I'$, and text, $T'$, with different appearance but identical semantic meaning. These can then be used to create three new image-text pairs for VLM training: $(I, T'), (I', T)$ and $(I', T')$. $I'$ is generated using the same random-crop augmentation used for simCLR image self-

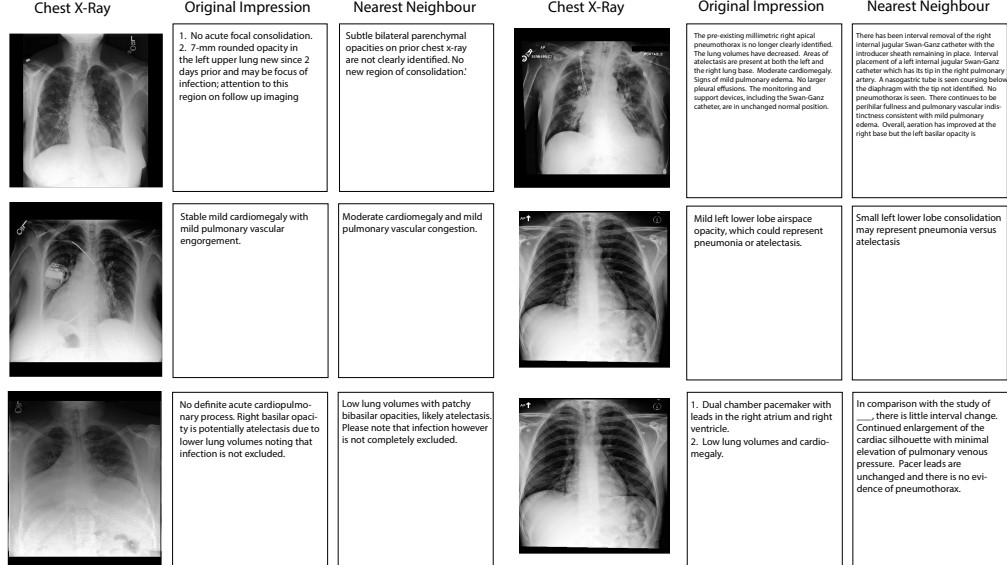

**Figure 6:** Example report nearest-neighbours found via the DeCLIP method using the model trained on 100% of the data. The text encoder finds nearest neighbour reports with the same semantic information as the original impressions, said in different words.

supervision described above. $T'$ is generated using the EDA method proposed by (Wei and Zou, 2019). This is a sentence-level simple text augmentation method which randomly: (a) selects $n_{syn}$ words from the text and replaces them with synonyms; (b) selects $n_{ins}$ words and inserts synonyms of them at random positions; (c) randomly swaps the positions of $n_{swap}$ word pairs (d) randomly deletes words with $p_{del}$. For a sentence of $W$ words, we use $n_{syn}, n_{ins}, n_{swap} = 0.1 \times W$ and $p_{del} = 0.1$. This is done using the official implementation of the method[4].

**Nearest-Neighbour Text Supervision:** Augmentation provides one method of generating new semantically-similar alternatives to the text and images. Another approach is to sample other images or reports from the training dataset which are projected into similar points in latent space by the dual encoder. We do this for text reports from the dataset. Practically, this is done by storing the last $N$ text embeddings produced by the text encoder in a first-in, first-out (FIFO) queue. Then for each report embedded, the nearest neighbour, $\mathbf{t}_{NN}$, can be found by measuring the maximum similarity between the report's global embedding vector and all elements in the queue. This can then be used to create two new positive pairs for VLM training, using the original and random-cropped images. Some examples of nearest neighbours found via this method are shown in Figure 6.

**Combined Loss Function:** The result is three additional components to the overall VLM loss function - the losses from text and image self-supervision, $\mathcal{L}_{TSS}$, $\mathcal{L}_{ISS}$, the mean loss from the three additional image-text pairs introduced using multi-view supervision, $\mathcal{L}_{MVS}$ and from the two additional pairs generated by nearest neighbour supervision, $\mathcal{L}_{NN}$. These

---

4. https://github.com/jasonwei20/eda_nlp

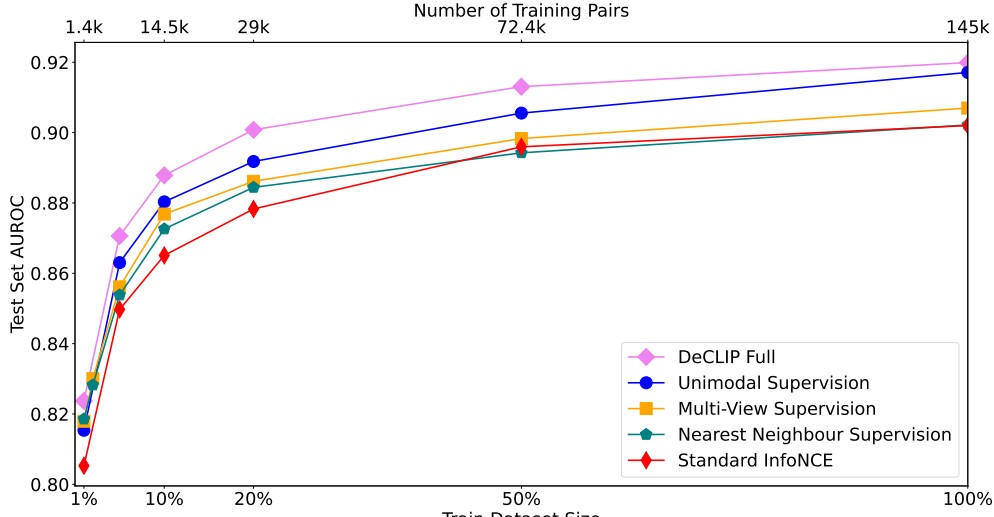

**Figure 7:** Ablation study for each individual component of the DeCLIP framework. In each case, depending on which component is being studied, two of $(\alpha, \beta, \gamma)$ from Equation 5 are set to 0, while the other equals 0.5.

are combined as follows:

$$\mathcal{L} = (1 - \alpha - \beta - \gamma)\mathcal{L}_{orig.} + \frac{\alpha}{2}(\mathcal{L}_{TSS} + \mathcal{L}_{ISS}) + \beta\mathcal{L}_{MVS} + \gamma\mathcal{L}_{NN}. \tag{5}$$

Here, $\mathcal{L}_{orig.}$ is the contrastive loss function for the original image-text pair. This can be either InfoNCE, the local component of GLoRIA, or a combination of the two. In the experiments in this paper we use $(\alpha, \beta, \gamma) = 0.2$.

**Ablation Study:** Figure 7 shows an ablation study where each of these components are studied in isolation, by setting two of $\alpha, \beta$ and $\gamma$ to zero and the other to 0.5, corresponding to the component being tested.

### C.1. Qualitative Retrieval Results

Figure 7 shows text-to-image retrieval examples for the best performing model, using the DeCLIP framework with a combined global and local loss. Each report is embedded and the global similarity with all images in the test dataset is calculated. Along with the query report and its associated image, we show the the retrievals at $k = 1, 2$, as well as the least similar image. From these results one can see that the trained model performs recall based on key conditions mentioned in the report, with almost all of the matching images having the same conditions present as those mentioned in the query.

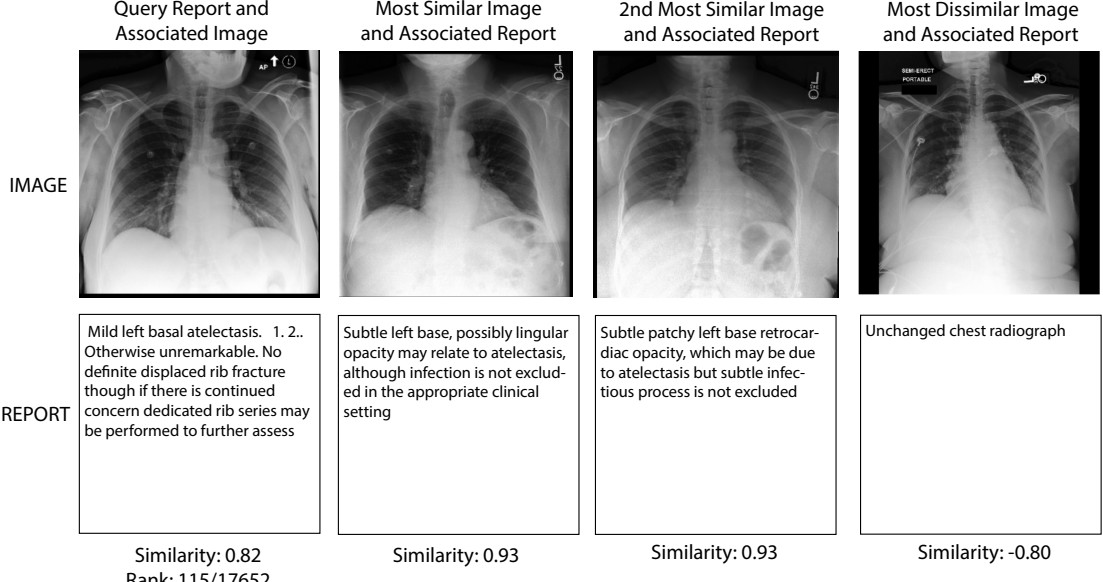

(a) Example query report demonstrating atelectasis. Note that first and second most similar images also exhibit atelectasis in the left lung base (as mentioned in the corresponding reports).

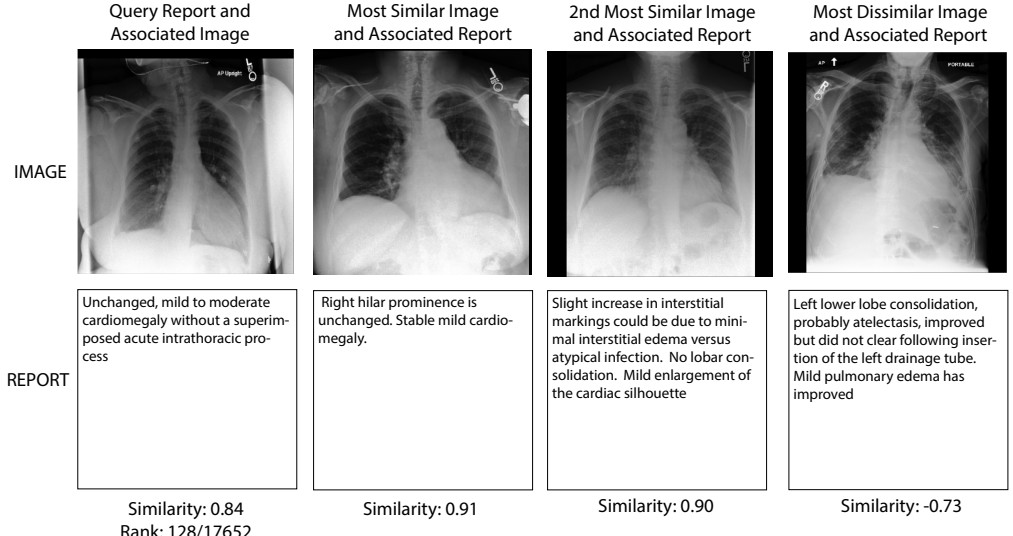

(b) Example report with mild cardiomegaly (enlarged heart). Again, the retrieved examples also exhibit mild cardiomegaly.

**Figure 7:** Example text-to-image retrieval results for the final model. All images in the test dataset are ranked based on similarity to the query report. Based on this criteria, the first, second and least most similar image-text pairs are shown, along with the original report and it's associated image.

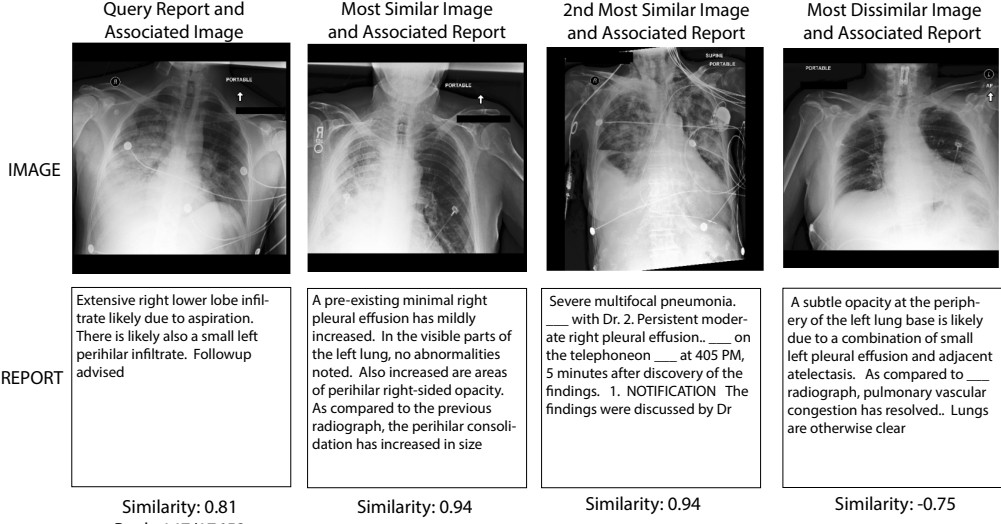

(*c*) Example query report with infiltration of the right lung.

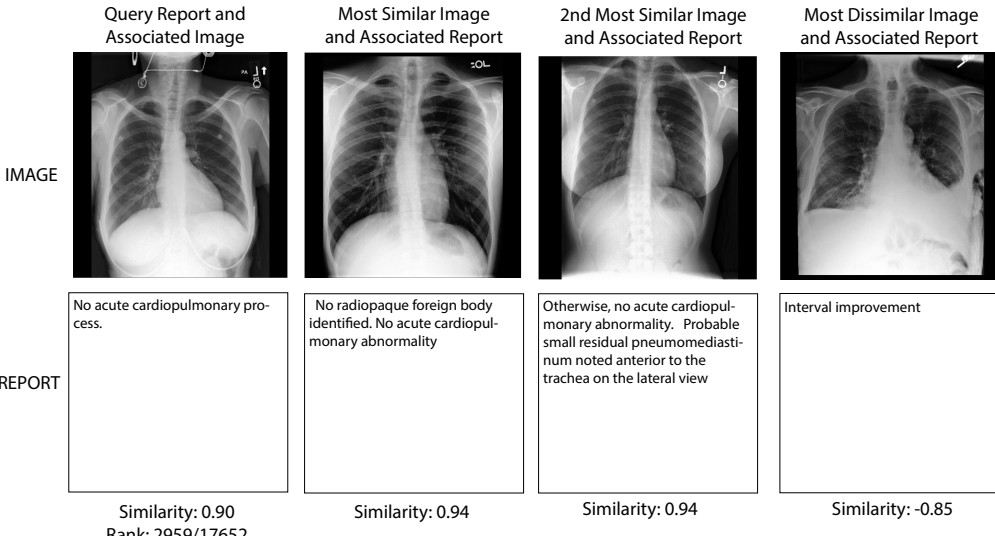

(*d*) An example query reporting no findings. Since this is true of a large subset of the dataset, the retrieval rank is fairly low here (2,959[th] out of 17,652 candidate images), despite high similarity.

**Figure 7:** (continued) Example text-to-image retrieval results for the final model. All images in the test dataset are ranked based on similarity to the query report. Based on this criteria, the first, second and least most similar image-text pairs are shown, along with the original report and it's associated image.

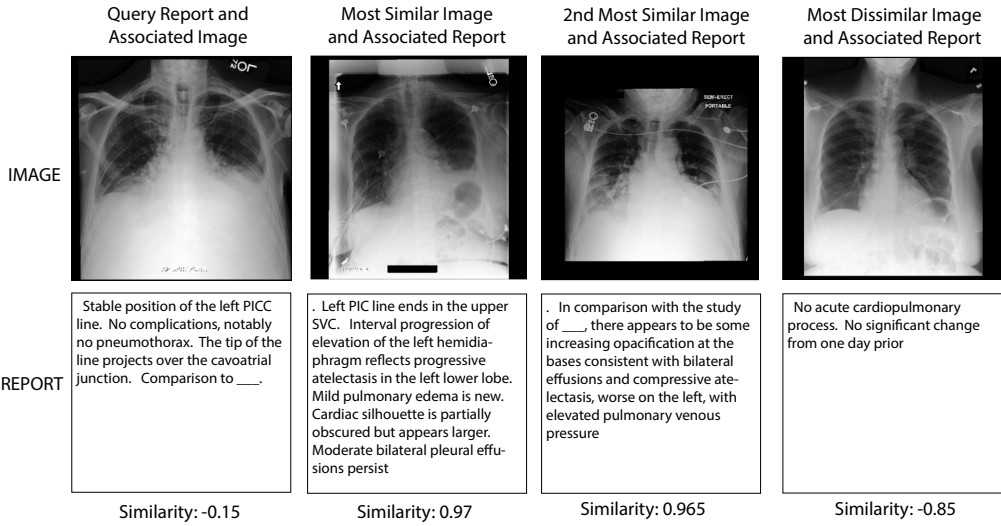

|  | Query Report and Associated Image | Most Similar Image and Associated Report | 2nd Most Similar Image and Associated Report | Most Dissimilar Image and Associated Report |

(e) Example failure case. The original image has a very low similarity, perhaps because the PICC line is very challenging to see in this case.

**Figure 7:** (continued) Example text-to-image retrieval results for the final model. All images in the test dataset are ranked based on similarity to the query report. Based on this criteria, the first, second and least most similar image-text pairs are shown, along with the original report and it's associated image.

## Appendix D. Downstream Task Evaluation Details

### D.1. Zero-shot Classification and Linear Probing

We evaluate our models at zero-shot classification on our test-split of MIMIC-CXR. Specifically we attempt to classify 12 conditions labelled by the CheXpert labeller (Irvin et al., 2019): Cardiomegaly, Atelectasis, Lung Opacity, Pleural Effusion, Edema, Pneumonia, Pneumothorax, Consolidation, Fracture, Enlarged Cardiomediastinum, Support Devices, Lung Lesion. We leave out the 'Pleural Other' and 'No Finding' classes, since these are difficult to write an all-encompassing prompt for - 'No Finding' since it indicates the absence of all conditions, rather than the presence of a specific one and 'Pleural Other' since it is very rare and can refer to a wide range of pleural disorders. Note that we treat all 'uncertain' cases, i.e. where the condition is not mentioned in the report, as assumed negatives.

**Zero-shot Classification Method:** For each class X, we embed two phrases using the trained text encoder; "X remains visible" and "There is no evidence of X", giving prompt embedding vectors $\mathbf{p}_X^+$ and $\mathbf{p}_X^-$. Then for a given report-image pair, the cosine similarity between the (normalized) prompt embeddings and image global embedding vectors is measured, $\{\mathbf{v}_i.\mathbf{p}_X^+, \mathbf{v}_i.\mathbf{p}_X^-\}$. The zero-shot prediction for each class is then measured by softmax($\mathbf{v}_i.\mathbf{p}_X^+, \mathbf{v}_i.\mathbf{p}_X^-$).

**Linear Probing Method:** As an additional method to measure the quality of the image encoder's features, we perform linear probing on the test dataset. This is done by 5-fold cross validation, performing class-weighted logistic regression (since the dataset is very im-

balanced) on 80% of the test data and testing on the remaining 20% of data. We report the mean balanced accuracy across all classes, averaged over all 5 folds.

The classification of both these methods is shown at the label-level in Table 4.

| Training Pairs (%) | Model Name | Balanced Accuracy (Zero-shot/Linear Probe, %) | | | | | | | |
|---|---|---|---|---|---|---|---|---|---|
| | | Cardiomegaly | Atelectasis | Lung Opacity | Pleural Effusion | Edema | Pneumonia | Pneumothorax | Consolidation |
| 1 | CLIP | 45.1/68.7 | 61.5/68.2 | 58.6/70.2 | 75.8/78.3 | 56.0/61.1 | 56.7/54.0 | 54.4/61.4 | 56.2/65.1 |
| | Ours | 68.0/70.6 | 67.9/69.2 | 70.9/71.9 | 76.5/80.7 | 61.7/61.3 | 51.0/55.1 | 55.3/60.6 | 65.4/66.4 |
| 5 | CLIP | 65.9/69.7 | 65.3/68.5 | 70.2/72.1 | 74.2/79.7 | 59.6/63.3 | 54.1/59.2 | 55.7/64.0 | 62.1/66.1 |
| | Ours | 70.1/72.9 | 69.4/71.9 | 71.8/74.2 | 76.7/81.8 | 62.5/63.1 | 51.5/57.3 | 56.9/64.3 | 66.2/67.4 |
| 10 | CLIP | 67.8/70.5 | 66.7/70.5 | 66.7/72.3 | 77.1/80.0 | 56.6/62.9 | 53.4/58.6 | 57.5/63.5 | 60.9/66.2 |
| | Ours | 69.2/72.8 | 69.1/73.3 | 72.0/74.9 | 76.6/81.7 | 62.4/62.9 | 52.8/57.4 | 59.5/66.0 | 66.7/68.6 |
| 20 | CLIP | 68.1/72.2 | 68.4/70.4 | 70.5/72.8 | 73.7/80.7 | 62.4/65.5 | 53.2/59.2 | 58.7/63.1 | 66.1/66.7 |
| | Ours | 70.2/73.6 | 68.7/73.3 | 72.9/75.7 | 76.4/82.2 | 63.8/65.4 | 54.4/59.0 | 60.8/66.8 | 66.6/68.8 |
| 50 | CLIP | 69.3/72.9 | 65.0/72.7 | 70.8/75.1 | 71.6/81.3 | 59.6/65.6 | 50.5/59.9 | 58.6/64.9 | 66.6/68.0 |
| | Ours | 70.1/74.1 | 69.6/74.3 | 73.6/76.3 | 76.8/82.5 | 63.8/64.4 | 57.0/61.8 | 63.8/66.7 | 67.5/69.4 |
| 100 | CLIP | 69.9/73.7 | 68.2/73.4 | 71.8/75.5 | 74.7/81.3 | 63.9/65.2 | 54.6/60.5 | 49.8/67.6 | 66.7/68.1 |
| | BioVIL | 70.2/73.8 | 69.6/73.2 | 74.4/74.7 | 72.5/80.4 | 63.0/65.2 | 52.1/60.9 | 60.1/65.2 | 66.2/68.7 |
| | Ours | 70.1/74.7 | 69.5/74.6 | 74.1/76.5 | 75.0/83.0 | 63.3/66.1 | 57.3/60.7 | 63.7/69.2 | 67.2/70.0 |

| Training Pairs (%) | Model Name | Balanced Accuracy (Zero-shot/Linear Probe, %) | | | | |
|---|---|---|---|---|---|---|
| | | Fracture | Enlarged Cardiomediastinum | Support Devices | Lung Lesion | **Mean** |
| 1 | CLIP | 75.1/76.4 | 54.1/59.7 | 61.3/68.0 | 74.7/77.7 | 60.8/ 67.4 |
| | Ours | 76.7/79.8 | 59.9/63.5 | 64.3/69.8 | 70.5/77.8 | 65.7/ 68.9 |
| 5 | CLIP | 77.5/79.3 | 57.5/61.5 | 62.3/71.5 | 68.0/79.3 | 64.4/ 69.5 |
| | Ours | 79.1/81.4 | 60.5/65.0 | 62.7/74.3 | 67.1/80.3 | 66.2/ 71.2 |
| 10 | CLIP | 78.1/80.3 | 58.9/61.5 | 62.2/72.1 | 69.6/79.9 | 64.6/ 69.9 |
| | Ours | 79.2/82.1 | 61.6/66.1 | 61.9/75.4 | 66.4/81.6 | 66.5/ 71.9 |
| 20 | CLIP | 80.4/81.1 | 60.4/63.3 | 68.4/74.2 | 70.4/80.8 | 66.7/ 70.8 |
| | Ours | 79.0/82.6 | 62.1/66.5 | 57.7/76.8 | 68.2/82.8 | 66.7/ 72.8 |
| 50 | CLIP | 79.4/81.7 | 61.4/63.4 | 63.3/76.4 | 72.4/81.5 | 65.7/ 71.9 |
| | Ours | 80.8/83.2 | 62.4/67.0 | 68.0/79.4 | 70.7/84.4 | 68.7/ 73.6 |
| 100 | CLIP | 78.4/82.1 | 62.2/64.5 | 64.4/76.9 | 58.7/82.7 | 65.3/ 72.6 |
| | BioVIL | 79.4/82.4 | 62.0/66.1 | 70.1/78.7 | 68.6/83.9 | 67.3/ 72.9 |
| | Ours | 79.9/83.7 | 62.9/68.3 | 68.8/81.0 | 70.0/85.6 | 68.5/ 74.5 |

**Table 4:** Full class level results for the zero-shot and linear probe classification experiments using CheXpert labels on the test set. Balanced accuracy scores are reported as 'Zero-shot accuracy/Linear Probe accuracy' for each task.

