# OpenReview forum: "Vision-Language Modelling For Radiological Imaging and Reports In The Low Data Regime"
_MIDL.io/2023/Conference — MIDL 2023 Oral_

### Official Review · Reviewer_GPeh · 2023-02-03

**Confidence:** 3
**Preliminary Rating:** 4

**Summary:**

The authors present vision language models for text to image retrieval using chest X ray images (CXR) and associated radiological reports.  Methods to improve performance, particularly in limited data settings are explored.  These include use of pretrained unimodal models via self supervision, the use of local and global contrastive loss functions and different types of additional supervision during training.

**Strengths:**

The paper is well written in clear English and explanations are good.  The topic is interesting and they demonstrate good performance with clear experiments that illustrate the benefits of particular approaches.

**Weaknesses:**

Some clarifications are required in certain areas of the text.  The metric of performance is not clearly explained, nor is any example of input/output visualized in a figure in the text.  Although the authors compare several (related) approaches and show performance for each one, it seems that most of these methods have a similar increase in performance as the size of the training dataset grows (so while they have found a superior method it is not specifically optimal with small training datasets)

**Deanonymize Review:**

no

**Detailed Comments:**

Please clarify explicitly what the performance metric is - text-to-image retrieval - does it mean that you input text and output a score for each image, so the ROC is made using scores of the correct images?

Some qualitative example figures would be nice showing the text input and the CXRs retrieved (with scores and true text)

Please note how the data was split (random? single split per patient?) and preferably make the split public for future comparison works.

On the x axis of the plots it would be useful to note the numbers of images in the training sets used (not just the %).

In figure legends use the same text convention and preferably the same line colour/type if the same information is plotted in two different plots (4(a) and 4(b))

It would be useful to have a comparison of training/inference times for different methods


**Paper Type:**

both

**Questions To Address In The Rebuttal:**

The points I have noted above are mostly minor issues so I would like the authors to address all of those unless there is some reason not to do so.  If the figure cannot be included in the main text it could be added in the appendix.

---

### Official Review · Reviewer_Gq41 · 2023-02-05

**Confidence:** 4
**Preliminary Rating:** 4
**Recommendation:** Poster

**Summary:**

The purpose of this research is to examine ways to train vision-language models (VLMs) in low-data scenarios. The authors examine techniques including adapting pre-trained models, utilizing contrastive loss functions (global and local), and providing extra supervision in VLM training. The study evaluates the techniques using chest X-rays and their radiology reports as a benchmark, and concludes that with proper pre-training and supervision, a dual-encoder VLM trained on a small number of image-text pairs can perform similarly to one trained on a large number. The authors also offer advice for researchers seeking to train VLMs in limited medical imaging data.

**Strengths:**


- Addresses a hot topic in the field of AI and computer vision by exploring methods to train vision-language models (VLMs) in data-scarcity environments.
- The authors evaluate the effectiveness of various state-of-the-art strategies, such as adapting pre-trained models, using contrastive loss functions, and providing extra supervision during training.
- The promising results show that with appropriate pre-training and supervision, VLMs trained on just a few thousand image-text pairs can perform comparably to those trained on hundreds of thousands of pairs.
- This approach has the potential to extend VLMs successfully to more image modalities, such as MRI and CT scans, even with scarce data.$^1$
- The experiments of the study are important and valuable, as there are few works that address this challenge and the results could have a significant impact on the field$^{2,3}$.

[1] Chambon et al. "Adapting Pretrained Vision-Language Foundational Models to Medical Imaging Domains". ArXiv. 2022

[2] Qin et al. "Medical Image Understanding with Pretrained Vision Language Models: A Comprehensive Study". ArXiV 2022

[3] Zhang et al. "Contrastive Learning of Medical Visual Representations from Paired Images and Text". ArXiC. 2022

**Weaknesses:**


The concern of generalization and potential biases in the experiments is a significant issue in the field of machine learning. When only a single dataset is used for both training and testing, it can lead to overfitting and inaccurate results due to the presence of similar data in both sets. The "unseen" data in the test set may not truly be unseen as it originates from the same distribution as the training data, leading to potential biases.

In this paper, the authors use the MIMIC-CXR dataset and split it into training, validation, and test sets. However, it is unclear if the test data is truly "unseen" as it comes from the same distribution as the rest of the data. This raises questions about the generalizability of the results and the validity of the conclusions drawn from the experiments. To mitigate these concerns, it would be more appropriate to use multiple datasets or employ techniques such as cross-validation to better evaluate the generalization performance of the model.

This highlights the importance of ensuring that the results of machine learning models are free from biases and accurately represent their generalization performance. Using multiple datasets and employing techniques such as cross-validation can help to mitigate these issues and improve the reliability of the results.

**Deanonymize Review:**

yes

**Detailed Comments:**

Given the number of reviews and the sort of rebuttal period, I prefer to avoid minor and detailed comments.  I hope that both the authors and the AC are in agreement with this. If you need clarification on any point, please let me know.

**Paper Type:**

validation/application paper

**Questions To Address In The Rebuttal:**


- Consider conducting inference on other datasets to further demonstrate the generalizability of the model and to stay up-to-date with the state-of-the-art studies that often employ multiple datasets$^4$. It could be nice to create a docker for inference to allow users to run the model on their own datasets with ease.
- Release the code anonymously to allow others to build upon the work and make it easier for the community to evaluate and reproduce the results.
- As the authors point out, it would be nice to enhance the representation of diverse cases by expanding the data selection criteria beyond just X-rays with a written "Impression" section. do you have some previous results on this?
- It would be valuable to have clarification from the authors on the pre-training process employed, including the amount of data used. This information would provide a deeper understanding of the results and allow for better comparison with other related works.
- It would be great to hear the authors' perspective on the pre-training process, particularly with regard to the results obtained when using ImageNet as the basis for pre-training the image encoder since is proved useless in numerous applications $^5$. It would also be interesting to understand the reasoning behind pretraining the text encoder within the medical domain while leaving the image encoder un-pretrained. Could the authors share their insights on this aspect of the study and its impact on the results?
-  Are there any considerations for balancing the contribution of these losses, and how do they impact the final results?
- I noticed in the "Evaluation on Downstream Tasks" section, you seem to be referring to Figure 5 instead of Figure 3. Is this an error or am I missing something?

[4] Wang et al. "MedCLIP: Contrastive Learning from Unpaired Medical Images and Text". ArXiV. 2022

[5] Raghu et al. "Transfusion: Understanding transfer learning for medical imaging". NeurIPS. 2019

---

### Official Review · Reviewer_zT1D · 2023-02-06

**Confidence:** 3
**Preliminary Rating:** 5
**Recommendation:** Oral

**Summary:**

This paper focuses on the Vision-Language Modelling for Radiological Imaging and Reports in the Low Data Regime.

It compares three different kinds of methods to improve low-data performance.

The experiments are conducted on a dataset with paired chest X-rays and radiological reports.

The conclusion is that training vision-language models on other medical imaging modalities when training data is scarce is beneficial.

**Strengths:**

- The paper is well-written and easy to follow with a clear definition of validation paper.
- The experiments are extensive and easy to see the comparison of different methods under different settings with line charts.
- The conclusion is clear and useful for future researchers.

**Weaknesses:**

- The experiments are only conducted on one dataset.
- The content in Fig. 1 is too small to see clearly.
- The results in the line charts conducted once or the average of multiple tries?
- I notice a footnote on Page 8: "The paper does not report a train-test split, thus pairs in our test set may be used in training this model." I don't think it is fair that train and test sets are in the same pool.

**Deanonymize Review:**

no

**Paper Type:**

validation/application paper

**Questions To Address In The Rebuttal:**

- The experiments are only conducted on one dataset.
- The content in Fig. 1 is too small to see clearly.
- The results in the line charts conducted once or the average of multiple tries?
- I notice a footnote on Page 8: "The paper does not report a train-test split, thus pairs in our test set may be used in training this model." I don't think it is fair that train and test sets are in the same pool.

---

### Meta-Review · Area_Chair_1fjx · 2023-02-24

**Recommendation:** Accept (Poster)
**Confidence:** 4

**Metareview:**

This paper explores methods to improve the performance of medical vision-language models (VLMs) with contrastive frameworks such as CLIP on limited training data, using chest X-rays and radiological reports as a benchmark. The proposed methods significantly improve retrieval and outperform existing benchmarks in downstream classification tasks. The reviewers have given relatively good comments on this work and the authors have robustly responded to all the comments from the reviewers.
Therefore an acceptance recommendation can be given here.